# Formation of Diffusion Layer on Ti-6Al-4V Alloy during Longtime Friction with Al

**Ruoyu Liu** [1,2,†] , **Chuanbing Huang** [1,†] , **Lingzhong Du** [1] , **Hao Lan** [1] , **Shige Fang** [1] , **Huifeng Zhang** [1] **and Weigang Zhang** [1,2,*]

1 State Key Laboratory of Multi-phase Complex Systems, Institute of Process Engineering, Chinese Academy of Sciences, Beijing 100190, China; liuruoyu14@mails.ucas.ac.cn (R.L.); cbhuang@ipe.ac.cn (C.H.); lzdu@ipe.ac.cn (L.D.); hlan@ipe.ac.cn (H.L.); shgfang@ipe.ac.cn (S.F.); hfzhang@ipe.ac.cn (H.Z.)
2 School of Chemical Engineering, University of Chinese Academy of Sciences, Beijing 100049, China
* Correspondence: wgzhang@ipe.ac.cn
† Contributed to this work equally.

**Abstract:** Elements' diffusion between Ti-6Al-4V alloy and Al in the process of sliding was investigated from 400 °C to 600 °C. The results showed that the atoms were diffused at the Ti-6Al-4V/Al interface. When sliding at 400 °C and 500 °C, no intermetallic compound was detected on the surface of Ti-6Al-4V alloy. When sliding at 600 °C, intermetallic $TiAl_3$ was formed and grew on the surface of Ti-6Al-4V alloy. The diffusion thickness of Al atoms on the surface of Ti-6Al-4V alloy increased with increasing time and temperature. The diffusion kinetic equation of Al atoms on the surface of Ti-6Al-4V alloy was successfully established. The diffusion activation energy of Al atoms in the sliding process was calculated to be 28.22 kJ·mol$^{-1}$ and the dynamic index $n$ was 0.5. The diffusion growth of Al atoms was controlled by normal parabolic law with time.

**Keywords:** Ti-6Al-4V; Al; sliding; diffusion

## 1. Introduction

Using frictional technology to improve the surface properties of materials is a common method in various manufacturing techniques, such as automobiles and aircraft engines [1–4]. The frictional technology of Ti-6Al-4V alloy and Al materials is widely used in friction stir welding and blade protection [5]. For example, the strength and high temperature properties of Ti-6Al-4V alloy are improved by friction stir welding [6]. In the field of aero engines, the wear properties of titanium blades are improved by sliding with Al materials [7–9]. Hence, the life of the Ti-blade is increased. However, the jointing of Ti-to-Al materials is unclear in the sliding process owing to significant differences in the physical properties of Ti and Al alloys.

Stringer and Marshall investigated the incursion rates in the process of Ti-6Al-4V sliding with AlSi-BN [10]. The results showed that Al is easily transferred to the surface of Ti-6Al-4V alloy at low incursion rates. Xue and Gao investigated the linear speed in the process of Ti-6Al-4V sliding with Al-BN [11], showing that the intermetallic compound is formed at the Ti-6Al-4V/Al interface at 150 m/s. Meanwhile, the thickness of the deformed layer on the surface of Ti-6Al-4V alloy becomes thick at high linear speeds. It is clear that Ti and Al atoms are diffused at the Ti-6Al-4V/Al interface.

Vanloo investigated the diffusion behavior between Ti-6Al-4V alloy and Al alloy at different temperatures [12] and found that the diffusion thickness of Al atoms follows Fick's second law. Zhang investigated the combustion synthesis of of $TiAl_3$ [13], and reported that the intermetallic compound $TiAl_3$ is formed at a temperature from 520 °C to 630 °C; meanwhile, the diffusion growth of $TiAl_3$ reaction layer is controlled by parabolic growth., and the diffusion dynamics equation of $TiAl_3$ was

established. Choi investigated the formation mechanism of TiAl and Ti$_3$Al [14], and showed that their formation was attributed to the Al diffusion from the TiAl$_3$. However, no research has focused on the diffusion of elements between Ti-6Al-4V alloy and Al in the process of sliding at different temperatures.

In this study, the details of elements' diffusion between Al and Ti-6Al-4V alloy in the sliding process were investigated at different temperatures. The diffusion kinetic equation of Al atoms at the Ti-6Al-4V/Al interface was investigated.

## 2. Material and Methods

### 2.1. Materials and Sample Preparation

Ti–6Al–4V alloy samples with the following chemical composition (wt.%) were studied: 6.6 Al, 4.0 V and balance titanium. The surface of Ti–6Al–4V alloy was cut into a circle with a 5-mm radius circle using a wire electrode cutting method, and the height of Ti–6Al–4V alloy was approximately 10 mm. Next, the surface of Ti–6Al–4V alloy was abraded with sandpaper ($R_a$ 0.5 μm) to keep the final surface roughness at approximately $R_a$ 0.5 μm. Finally, Ti–6Al–4V alloy was cleaned ultrasonically with absolute ethanol, dried in warm air, and then stored in a dryer until the test. The counterpart was made of Al (99.9%) and was a rectangle of 25 mm × 25 mm × 6 mm. The surface of the counterpart was abraded with sand paper to keep the final surface roughness value at approximately $R_a$ 0.5 μm. Finally, the counterpart was cleaned with absolute ethanol and then dried in warm air before the test.

### 2.2. Methods

#### 2.2.1. Testing Methods

The sliding tests were conducted by a high-temperature friction and wear tester (CSM-THT, Switzerland). The testing was conducted at temperatures of 400 °C, 500 °C, and 600 °C. The heating rate was 10 °C/min from room temperature to the testing value. The sliding tests were conducted at a load of 3 N and a line speed of the center of 0.1884 m/s. Each of the tests was repeated twice. A diagram of the sliding test device is shown in Figure 1.

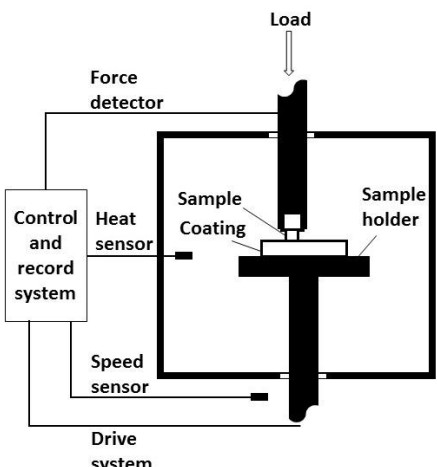

**Figure 1.** Schematic diagram of the high-temperature sliding device.

#### 2.2.2. Characterization

After sliding with Al at different temperatures, the cross-section of the worn surface of Ti-6Al-4V alloy was obtained with a wire electrode cutting equipment. The phase composition of cross-sectional Ti-6Al-4V alloy was determined using a Philips X'Pert Pro X-ray diffractometer (XRD, PANalytical, Eindhoven, The Netherlands) using Cu Kα radiation over an angular range from 30° to 50° at 40 kV, 40 mA. Backscattered electron (BSE) images of cross sectional Ti-6Al-4V alloy were obtained by using an

FEI Quanta 200 FEG scanning electron microscope (SEM, JEOL, Shoshima, Tokyo, Japan) equipped with an energy dispersive spectrometer (EDS) analysis system. Elemental contents of cross-sectional Ti-6Al-4V alloy were investigated by electron probe microanalysis 1720H (EPMA, Shimadzu, Kyoto, Japan). During data collection, an accelerating voltage of 20 kV and a beam current of 10 nA were used. The step of the electron beam during data collection was 0.1 μm.

## 3. Results and Discussion

### 3.1. Diffusion Behaviour on the Surface of Ti-6Al-4V Alloy at Different Temperatures

XRD patterns of cross-sectional Ti-6Al-4V alloy sliding at 400 °C, 500 °C, and 600 °C for 10 h are shown in Figure 2. The peak at 38.47° belongs to [111] of Al. The peak at 44.74° belongs to [200] of Al. The peak at 40.41° belongs to [101] of α-Ti. The peak at 35.3° belongs to [100] of α-Ti. The peak at 38.75° belongs to [110] of β-Ti. The peak at 39.25° belongs to [103] of TiAl3. As shown in Figure 2, Al and Ti are detected when sliding from 400 °C to 600 °C. Ti represents Ti-6Al-4V alloy; this means that Al is transferred to the surface of Ti-6Al-4V alloy sliding from 400 °C to 600 °C. When sliding at 600 °C, $TiAl_3$, Al, and Ti are detected, which means that $TiAl_3$ is formed when sliding at 600 °C.

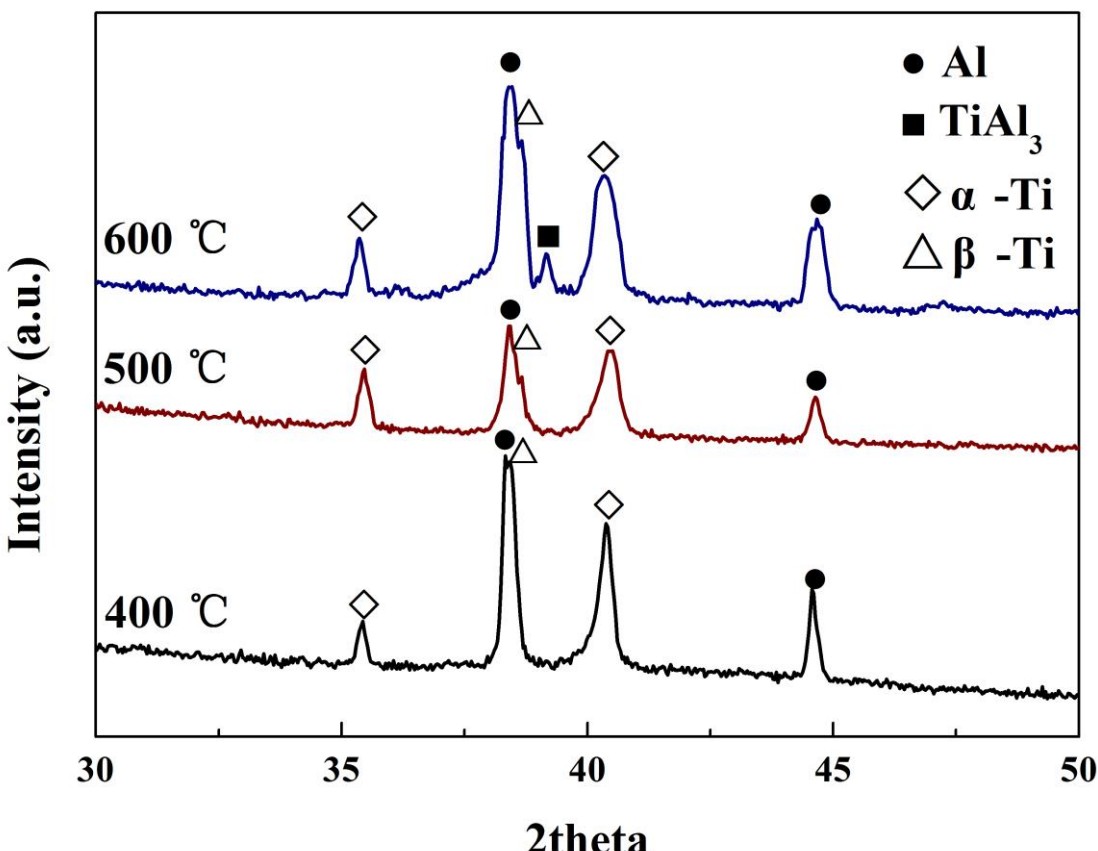

**Figure 2.** X-Ray diffraction (XRD) patterns of cross-sectional Ti-6Al-4V alloy sliding at 400 °C, 500 °C and 600 °C for 10 h.

BSE images of cross-sectional Ti-6Al-4V alloy sliding at 400 °C, 500 °C, and 600 °C for 10 h are shown in Figure 3a,c, where the black, the grey, and the dark grey represent Al, Ti-6Al-4V alloy, and $TiAl_3$, respectively. When sliding at 400 °C and 500 °C, Al and Ti-6Al-4V are detected and no intermetallic compound is formed at the Ti-6Al-4V/Al interface. When sliding at 600 °C, the intermetallic compound $TiAl_3$ is detected. The intermetallic compound $TiAl_3$ is formed at the

Ti-6Al-4V/Al interface. The results show that Al is transferred to the surface of Ti-6Al-4V alloy in the sliding process from 400 °C to 600 °C.

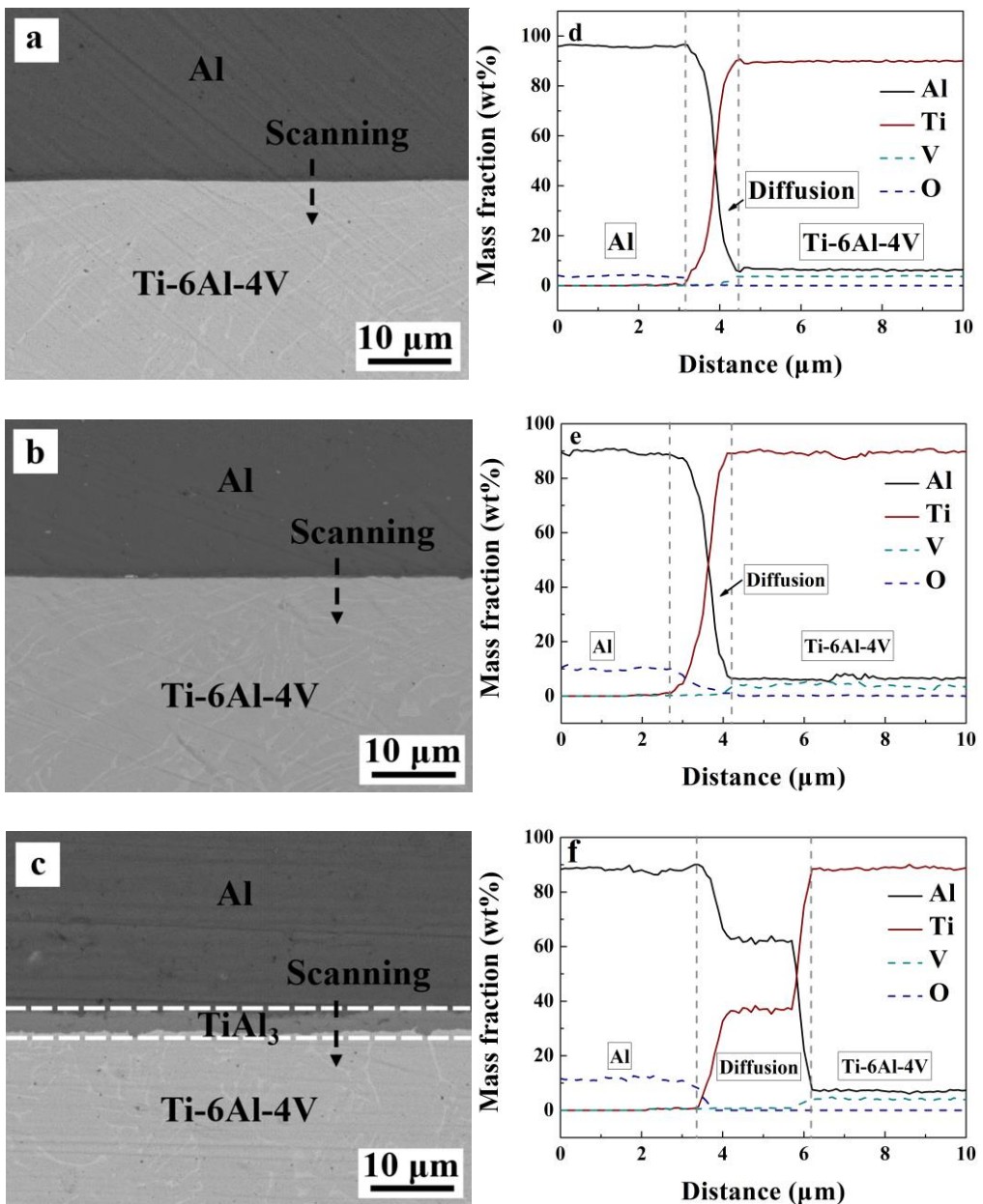

**Figure 3.** BSE images of cross-sectional Ti-6Al-4V alloy sliding at different temperatures for 10 h: (**a**) 400 °C; (**b**) 500 °C; (**c**) 600 °C. The results of EPMA of lines at the Ti-6Al-4V/Al interface: (**d**) 400 °C; (**e**) 500 °C; and (**f**) 600 °C.

The results of EPMA of lines at the Ti-6Al-4V/Al interface at 400 °C, 500 °C, and 600 °C for 10 h are shown in Figure 3d,f, where the black, red, blue, and green lines represent Al, Ti, O, and V, respectively. The ordinate value represents the mass fraction. As shown in Figure 3d, the content of Al is 90 wt.% and the content of Ti is almost 0 in the dark region. From the dark region to the grey region, the contents of Al and Ti present a sharp decrease and increase tendency, respectively. The content of O is less than 10 wt.% in the dark region. From the dark region to the grey region, the content of O decreases slowly. The content of V is about 0 in the dark region. From the dark region to the grey region, the content of V increases slowly. The results show that the atoms of Ti, Al, O, and V are

diffused at the Ti-6Al-4V/Al interface at 400 °C. The distance of Al contents from 90 wt.% to 6 wt.% is 1.1 µm. As shown in Figure 3e, from the dark region to the grey region, the contents of Al and Ti present a sharp decrease and increase tendency, respectively. From the dark region to the grey region, the contents of O and V present a decrease and increase tendency, respectively. The results show that the atoms of Ti, Al, O, and V are diffused at the Ti-6Al-4V/Al interface at 500 °C. The distance of Al contents from 90 wt.% to 6 wt.% is 1.6 µm. As shown in Figure 3f, from the dark region to the dark grey region, the contents of Al and Ti present a sharp decrease and increase tendency, respectively. In the dark grey region, the contents of Al and Ti are steady. From the dark grey region to the grey region, the contents of Al and Ti present a decrease and increase tendency, respectively. From the dark region to the dark grey region, the contents of O decrease. From the dark region to the grey region, the contents of V increase. The results show that the atoms of O are diffused at the TiAl$_3$/Al interface at 600 °C. The atoms of V are diffused at the TiAl$_3$/Ti-6Al-4V interface at 600 °C. The atoms of Ti and Al are diffused at the TiAl$_3$/Al and TiAl$_3$/Ti-6Al-4V interface at 600 °C. The distance of Al contents from 90 wt.% to 6 wt.% is 2.4 µm.

Due to the diffusion behavior of Al atoms, Al is transferred to the surface of Ti-6Al-4V alloy. The diffusion behavior of Ti, Al, V, and O atoms is affected by the temperature in the sliding process. The thickness diffusion of Al atoms is 1.1 µm at 400 °C, 1.6 µm at 500 °C, and 2.4 µm at 600 °C. The diffusion thickness of Al atoms increases with increasing temperature at the Ti-6Al-4V/Al interface.

### 3.2. Diffusion Kinetic of Al Atoms on the Surface of Ti-6Al-4V Alloy at Different Temperatures

The diffusion thicknesses of Al atoms at different temperatures and times are shown in Figure 4. From 400 °C to 600 °C, the diffusion thickness of Al atoms increases with increasing time. Meanwhile, the diffusion thickness of Al atoms increases with increasing temperature.

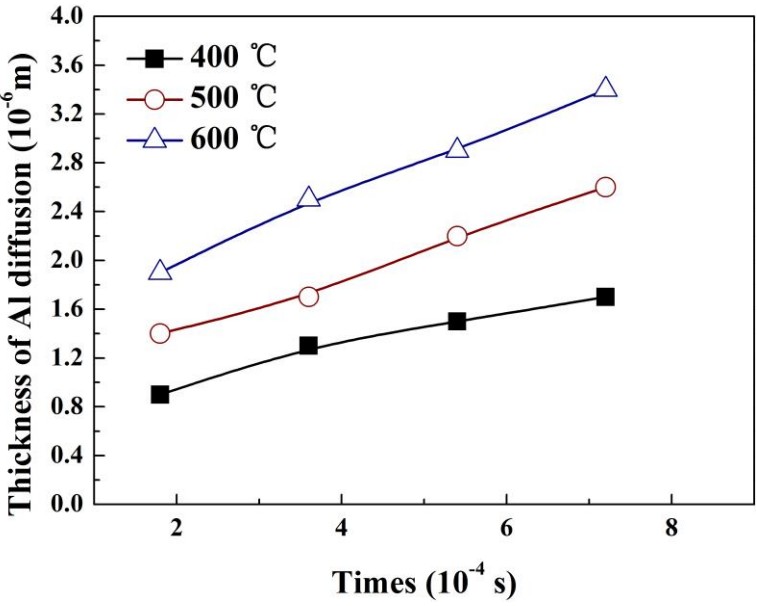

**Figure 4.** Diffusion thickness of Al atoms at different temperatures and times.

The influence of temperature and time on atomic diffusion can be described by the following empirical relationship [15]:

$$\Delta x = Dt^n \tag{1}$$

$$\ln(\Delta x) = n\ln(t) + \ln D \tag{2}$$

where $\Delta x$ is the diffusion thickness (m), $t$ is the diffusion time (s), $D$ is the growth constant of the phase, $n$ is the exponent. When $n$ is equal to 0.5, the diffusion growth of Al atoms is controlled by parabolic

growth kinetics; when $n$ is 1, the diffusion growth of Al atoms is dominated by linear growth. Usually, the value of $D$ can be assumed to follow an Arrhenius type as presented in Equation (3):

$$D = D_0 \exp(-\frac{Q}{RT}) \tag{3}$$

where $D_0$ is the pre-exponential factor ($m^2 \cdot s^{-1}$), $R$ is the gas constant ($8.31$ $J \cdot mol^{-1} \cdot K^{-1}$), $Q$ is activation energy for growth ($J \cdot mol^{-1}$), and $T$ is absolute temperature in Kelvin. According to Equations (2) and (3), the diffusion kinetic equation is described as Equation (4):

$$\ln(\Delta x) = \ln D_0 - \frac{Q}{RT} + n \ln t \tag{4}$$

The relationship between $\ln \Delta x$ and $\ln t$ is shown in Figure 5, where the slope of three lines is 0.46 at 400 °C, 45 at 500 °C, and 0.41 at 600 °C, respectively. The slope of the three lines is close to 0.5. Hence, the diffusion growth of Al atoms is controlled by parabolic growth kinetics and $n$ is 0.5.

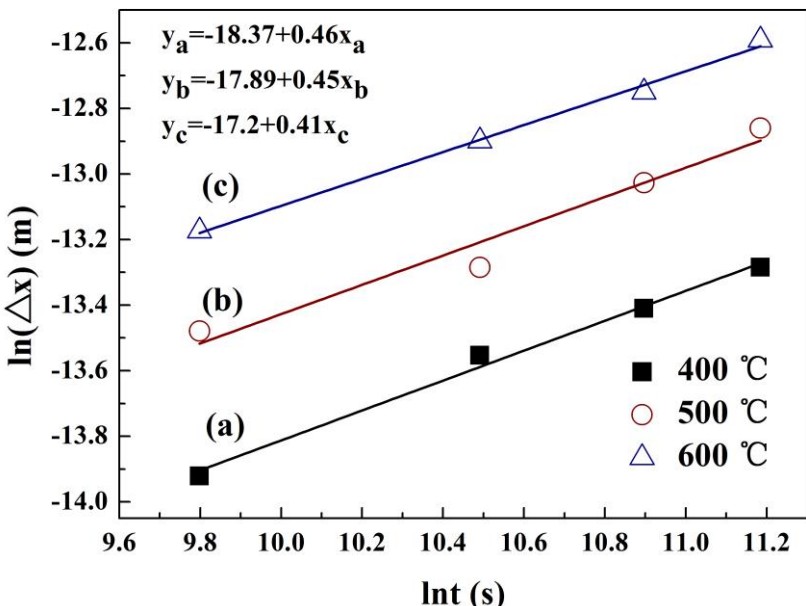

**Figure 5.** Relationship between the diffusion thickness of Al atoms and time at different temperatures.

The activation energy can be obtained from plots of $\ln D$ versus $1/T$. Such kinds of plots are depicted in Figure 6. It can be noted that the relationship between $\ln D$ and $1/T$ is presented as Equation (5):

$$\ln D = -13.38 - \frac{3395.58}{T} \tag{5}$$

According to Equations (4) and (5), the value of $Q/R$ is 3395.58, and $Q$ is 28.22 $kJ \cdot mol^{-1}$. As a result, the diffusion kinetic equation of Al atoms in the sliding process is presented as follows:

$$\Delta x = 0.03 \exp(\frac{-28.22 \times 10^3}{RT}) t^{\frac{1}{2}} \tag{6}$$

This equation shows that the diffusion thickness of Al atoms increases with increasing time and temperature.

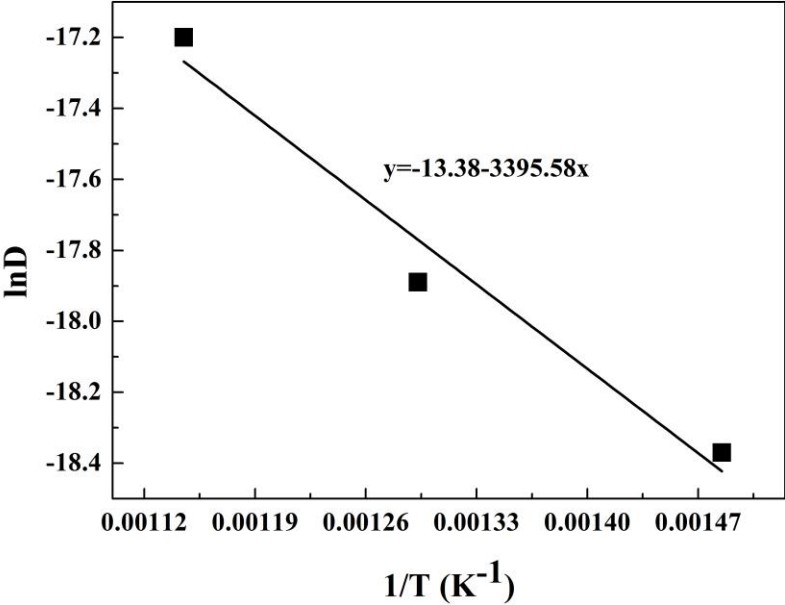

**Figure 6.** Relationship of ln*D* and 1/*T*.

## 4. Conclusions

The diffusion behavior of elements between Al and Ti-6Al-4V alloy in the sliding process at different temperatures was investigated. The diffusion kinetics of Al atoms at different temperatures in the sliding process were also examined. According to the sliding experiments between pure Al and Ti-6Al-4V alloy at different temperatures and times, the following important conclusions can be drawn:

(1)  When sliding at 400 °C and 500 °C, the diffusion of Ti, Al, O, and V atoms occurs at the Ti-6Al-4V/Al interface. When sliding at 600 °C, the intermetallic compound TiAl$_3$ is formed on the surface of Ti-6Al-4V alloy. Meanwhile, the diffusion of O atoms occurs at the TiAl$_3$/Al interface, and the diffusion of V atoms occurs at the TiAl$_3$/Ti-6Al-4V interface. The diffusion of Al and Ti atoms occurs at the TiAl$_3$/Ti-6Al-4V and TiAl$_3$/Al interfaces.

(2)  In the sliding process, the diffusion growth of Al atoms on the surface of Ti-6Al-4V alloy is controlled by parabolic growth kinetics.

(3)  In the sliding process, the diffusion thickness of Al atoms on the surface of Ti-6Al-4V alloy increases with increasing time and temperature.

**Author Contributions:** R.L. performed the data analyses and wrote the manuscript; C.H. helped perform the analysis with constructive discussions; H.Z. and S.F. helped document retrieval; L.D., H.L. and W.Z. performed the manuscript review.

**Funding:** This research was funded by [National Key R&D Program of China] grant number [2018YFC1902401], [National Natural Science Foundation of China] grant number [51671180] and [National Natural Science Foundation of China] grant number [51471159].

**Conflicts of Interest:** The authors declare no conflict of interest.

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
