# Peer review of "Formation of Diffusion Layer on Ti-6Al-4V Alloy during Longtime Friction with Al"

_metals, doi:10.3390/met9101107_

Round 1
Reviewer 1 Report
The topic of research is interesting whereby the authors have looked into the aspect of the diffusion of various elements to the surface during the sliding of the Titanium alloy with Aluminum. However, the manuscript needs a bit of improvement in terms of English language and from technical point of view. The suggestions are listed below:
1. Please revise the English language in terms of spelling mistakes/grammar as there are many instances in the manuscript where the language is not clear.
2. The introduction part should be re-written to make the language clear.
3. How are the authors sure that the Aluminum that they are seeing on the surface has been diffused or is it being transferred from the counterface pin during sliding. I am a bit confused here and the explanation is not very clear.
4. If the authors intended to study the diffusion of Al, then they should have used a pin of different material other than aluminum. Am, I missing a point here? Please explain.
5. How did the authors conduct the BSE SEM of the cross-section of the sample? Need to be explained more properly. How are they showing a thick Aluminum layer in the image? Is it of the diffused Al or the counterface? It is not clear in the image or in the explanation.
6. The authors should explain the above phenomenon clearly for the paper to be accepted for publication.
Author Response
We appreciate the editor and the reviewer very much for their positive and constructive comments on our manuscript. Revised portion are marked in red in the paper. Please see the attachment.

Reviewer 2 Report
Dear editor,
In the manuscript received, the authors study the influence of the diffusion layer generated during the friction between Ti6Al4V and Aluminum at high temperature. This manuscript may have some industrial interest, especially in burnishing applications or operations where there is friction between the part and the tool. Thus, it is accepted with minor revisions and the authors must to have into account to finish of improving the manuscript:
First of all, the state of the art has to be improved. During this section, the authors cite several papers per sentence. This implies a certain devirtualization and does not really contribute something. Therefore, they have to eliminate multiple references. For example: writing 1 or 2 phrases mentioning the issue claimed and why it deserves mentioned for each reference. (this happens along all the manuscript). On the other hand, it would be important also improve the state of the art including industrial applications of the results obtained here. For example in burnishing of FSW or friction-drilling process for two dissimilar materials which are regenerating new opportunities from and industrial point of view. Regarding section 2, in the line 52, the authors must include the grain size of the sand paper used. Finally, in section 3, it would be explained the numbers between “[ ]”. Besides, in the Figure 3, labels of the pictures have to increase the size because it is difficult to see them. Finally, the equations used in lines 132, 138, and 142 they should introduce the reference of it.Author Response
We appreciate the editor and the reviewer very much for their positive and constructive comments on our manuscript. Revised portion are marked in red in the paper. Please see the attachment.

Reviewer 3 Report
The authors of the above referenced manuscript describe a formation of diffusion layer on Ti-6Al-4V alloy during longtime friction with aluminum and study the formation of diffusion layer using X-ray Diffraction, EDS, SEM, and electron probe microanalysis. While the experiments are done well and results are new and potentially useful for some applications, the manuscript suffers from a few shortcomings that makes it unsuitable for publication in its present state.
General remark: an English should be improved – see for example
for example Abstract
The results showed that the atomic is atoms are diffused at the 10 Ti-6Al-4V/Al interface.
When sliding at 400 °C and 500 °C, no intermetallic is characterized detected on the 11 surface of Ti-6Al-4V alloy.
With the temperatures increased, the diffusion thickness of Al atoms on the surface of Ti-6Al-4V alloy increased. The diffusion kinetic equation of Al atoms on the surface of Ti-6Al-4V alloy was successfully established. With the sliding times and temperatures increased, the diffusion thickness of Al atoms increased. (repeated information in 1st and 3rd sentences)
With the temperatures increased, The diffusion thickness of Al atoms on the surface of Ti-6Al-4V alloy increase with the increase of the time and temperature.
and
Figure 2: XRD analysis: It is with poor quality. What is the error? It is better to provide a better refinement results. The intensity scale could be in arbitrary unit because the absolute values of intensities have no meaning.
The Ti-6Al-4V alloy is 2-phase material: a and b phases – so the Authors should add to the Fig. 2 also this XRD spectrum
Author Response

(The authors gave the same response as above.)

Round 2
Reviewer 1 Report
Thanks for replying and taking care of most of the comments effectively. However, the language to the comment #5 where the explanation is given about the cross-sectional analysis which is also included in the revised manuscript should be improved for the easy understanding of the reader.
Author Response
We appreciate the reviewer very much for the positive and constructive comments on our manuscript. Revised portion are marked in red in the paper. The responds to the reviewer’s comments are as flowing:
1.The language to the comment #5 where the explanation is given about the cross-sectional analysis which is also included in the revised manuscript should be improved for the easy understanding of the reader.
Answer: After sliding with Al, Ti-6Al-4V alloy was fixed in the resin. Then, Ti-6Al-4V alloy was cut by a wire electrode cutting equipment to obtain the cross section of the worn surface. To observe the morphology clearly, the cross section of the worn surface of Ti-6Al-4V alloy was polished with the sandpaper. The cross section of the worn surface of Ti-6Al-4V alloy was characterized by the BSE detector. As shown in Figure 3a, the black and the grey represents Al and Ti-6Al-4V alloy, respectively. Obviously, Al is transferred to the surface of Ti-6Al-4V alloy in the sliding process. As shown in Figure 3b, the atoms of Al are diffused at the Ti-6Al-4V/Al interface. Due to the diffusion behavior of Al atoms, Al is transferred to surface of Ti-6Al-4V alloy. The revised part in the manuscript is as follows:
Page3 Line74: After sliding with Al at different temperatures, the cross section of the worn surface of Ti-6Al-4V alloy is obtained by a wire electrode cutting equipment.
Page4 Line129: Due to the diffusion behavior of Al atoms, Al is transferred to surface of Ti-6Al-4V alloy.
If you have any questions, please do not hesitate to contact me.